# Optimizing the P balance: How do modern maize hybrids react to different starter fertilizers?

Thea Mi Weiß[1,2], Willmar L. Leiser[2], Alice-J. Reineke[3], Dongdong Li[4], Wenxin Liu[4], Volker Hahn[2], Tobias Würschum[1]*

1 Institute of Plant Breeding, Seed Science and Population Genetics, University of Hohenheim, Stuttgart, Germany, 2 State Plant Breeding Institute, University of Hohenheim, Stuttgart, Germany, 3 Institute of Agricultural Engineering in the Tropics and Subtropics, University of Hohenheim, Stuttgart, Germany, 4 Key Laboratory of Crop Heterosis and Utilization, the Ministry of Education, Key Laboratory of Crop Genetic Improvement, Beijing Municipality, National Maize Improvement Center, College of Agronomy and Biotechnology, China Agricultural University, Beijing, P.R. China

* tobias.wuerschum@uni-hohenheim.de

**Data Availability Statement:** All relevant data are within the paper and its Supporting information files.

## Abstract

Phosphorus (P) is an essential macronutrient for plants, but also a limited resource worldwide. Strict regulations for fertilizer applications in the European Union are a consequence of the negative environmental effects in case of improper use. Maize is typically grown with the application of P starter fertilizer, which, however, might be reduced or even omitted if suitable varieties were available. This study was performed with the 20 commercially most important maize hybrids in Germany evaluated in multi-location field trials with the aim to investigate the potential to breed for high-performing maize hybrids under reduced P starter fertilizer. At the core location, three starter fertilizers with either phosphate (triple superphosphate, TSP), ammonium nitrate (calcium ammonium nitrate, CAN), or a combination of ammonium and phosphate (diammonium phosphate, DAP) were evaluated relative to a control and traits from youth development to grain yield were assessed. Significant differences were mainly observed for the DAP starter fertilizer, which was also reflected in a yield increase of on average +0.67 t/ha (+5.34%) compared to the control. Correlations among the investigated traits varied with starter fertilizer, but the general trends remained. As expected, grain yield was negatively correlated with grain P concentration, likely due to a dilution effect. Importantly, the genotype-by-starter fertilizer interaction was always non-significant in the multi-location analysis. This indicates that best performing genotypes can be identified irrespective of the starter fertilizer. Taken together, our results provide valuable insights regarding the potential to reduce starter fertilizers in maize cultivation as well as for breeding maize for P efficiency under well-supplied conditions.

## 1 | Introduction

Phosphorus (P) is a globally limited reserve [1]. There are varying predictions of how long phosphate rock reserves will last, but most studies anticipate a time frame between 100 and

**Funding:** This work was funded by the Deutsche Forschungsgemeinschaft (DFG, German Research Foundation) – 328017493/GRK 2366 (Sino-German International Research Training Group AMAIZE-P). The funder had no role in study design, data collection and analysis, decision to publish, or preparation of the manuscript. TMW and AJR received a salary from the funder.

**Competing interests:** The authors have declared that no competing interests exist.

400 years [2]. Moreover, P reserves are unevenly distributed across the globe [3] and often contaminated with heavy metals [3, 4]. In living organisms, phosphorus always occurs in the form of phosphates ($P_i$) and plays an essential role as a component of the DNA, cell membranes, and coenzymes as well as in the energy transfer processes of cells. Thus, P is deemed one of the most important macronutrients for plants [5].

Maize (*Zea mays* L.) is one of the three major staple foods worldwide with a cultivated area of around 194 million hectares in 2018 [6], of which Germany grows more than 2.6 million ha, primarily for silage usage [7]. It is common agricultural practice since the 1980s in industrialized countries to apply starter fertilizers in maize cultivation, specifically combinations of ammonium and phosphate [8–10]. Germany, for instance, used over 90.8 Mt of P fertilizer in 2017 [11]. However, the known negative environmental effects [12, 13] due to an inappropriate use of fertilizers—in its worst form, the eutrophication of surface water by run-off and leaching into drainages or deeper soil layers–have gained increasing attention in the public perception and the shaping of agricultural policies [14, 15]. Recently, the Farm to Fork Strategy of the European Union was released, which aims for a reduction of nitrogen and phosphate fertilizers of 20% by 2030 [16]. In Germany, a new fertilizer ordinance has been introduced, dictating the documentation of nitrogen and phosphate fertilizer usage for the whole farm; fertilizer applications are restricted depending on the overall nutrient removal of each crop [17]. Several studies have shown that P in Europe is sufficiently to abundantly available on agricultural lands [18]. This holds specifically true for agricultural businesses that have a surplus of organic fertilizers available, namely livestock- and biogas-based farms [19, 20]. In many cases however, the abundant P is fixed by minerals and therefore not fully available for plants [21].

Taking these facts together, it now appears timely and reasonable to breed for an optimized use of phosphate fertilizers in maize in order to achieve an improved ecological footprint. Phosphate-use-efficiency (PUE) is classically defined either as high P concentrations of the harvested organs due to a higher P *uptake* of the roots or as increased yields per given unit P due to a better internal *utilization* of the available P [22–24]. In previous studies, it was shown that the traits early vigor, early-season plant height, flowering, and yield react to P deficiency in sorghum and can thus be considered as P-sensitive traits [23]. Conversely, starter fertilization in maize may lead to an increase in grain yield of 4.5% in comparison with broadcast fertilization [25].

Nevertheless, little is known about the reaction of modern maize hybrid varieties to different starter fertilizers. We therefore conducted a field trial with 20 modern maize hybrids evaluated at five locations within Germany under a control (Co) and three different starter fertilizers treatments, i.e. phosphate (triple superphosphate, TSP), ammonium nitrate (calcium ammonium nitrate, CAN), or a combination of ammonium and phosphate (diammonium phosphate, DAP). In particular, our objectives were to (i) assess the variation in the response to different P starter fertilizers in maize cultivation, (ii) evaluate the genotype-by-starter fertilizer interaction, (iii) identify high-yielding and P-stable maize hybrids, and (iv) draw conclusions for maize breeding.

## 2 | Material and methods

### 2.1 | Plant material

For this study, the 20 commercially most important maize hybrids in Germany were chosen. They belong to eight breeding companies and the vast majority represents the mid-early maturity group (FAO 200–270). All varieties are suited for grain or corn-cob-mix utilization and were harvested as grain maize. Moreover, all seeds were treated in the standard way of each company (S1 Table).

**Table 1. Description of the locations.**

| | Altitude [m ASL] | Ø Temp.* [˚C] | Ø Precip.* [mm] | Soil type | pH | $P_2O_5$ [mg/100g soil] | P [mg/100g soil] | Classification of P availability** |
|---|---|---|---|---|---|---|---|---|
| Hohenheim (HOH) | 402 | 10.6 | 857 | Silty Loam | 6.79 | 21.1 | 9.2 | D |
| Eckartsweier (EWE) | 142 | 11.7 | 783 | Clayey Loam | 6.54 | 19.2 | 7.7 | D |
| Dettingen (DET) | 561 | 9.1 | 661 | Clayey Loam | 7.33 | 52 | 20.5 | E |
| Einbeck (EIN) | 124 | 8.7 | 679 | Clayey Loam | 6.85 | 19.5 | 7.7 | D |
| Saerbeck (SAB) | 56 | 9.3 | 789 | Strongly Loamy Sand | 5.75 | 24.7 | 10.7 | D |

Including altitude, weather data (mean temperature, mean precipitation), soil type, and pH. According to the P status, the 'Classification of P availability' of the soils can range from A (very low) to E level (very high).

* Data for locations in Baden-Württemberg retrieved from <www.wetter-bw.de>, for locations outside of Baden-Württemberg retrieved from <climate-data.org>

** According to VDLUFA-P-content-classes (A = very low, E = very high) defined by the Association of German Agricultural Analytic and Research Institutes (Verband Deutscher Landwirtschaftlicher Untersuchungs- und Forschungsanstalten).

## 2.2 | Field trial

We applied four treatments, i.e. a control (Co: 0% N/ 0% P), and the three different starter fertilizers triple superphosphate (TSP: 0% N/ 20% P), calcium ammonium nitrate (CAN: 26% N/ 0% P), and diammonium phosphate (DAP: 18% N/ 20% P). The field design was laid out as an alpha-lattice (5×4) using the software CycdesigN [26]. Genotypes were replicated twice per starter fertilizer treatment and the trial was conducted at five different locations. Hohenheim served as core location with the control and all three different starter fertilizers. All other locations comprised the control and either TSP or DAP, resulting in total in three locations with TSP, three with DAP, and one with CAN (S2 Table). The weather data including soil temperatures in Hohenheim (S1 Fig) for the calendar year 2019 [27] was characterized by an extraordinary cold phase in May right after sowing the trial, which led to a delayed emergence. The altitude of the field locations ranged from 56 to 561 m above sea level, the average temperatures varied from 8.7 to 11.7˚C, and the average annual precipitation amounted to 661 to 857 mm in 2019. All locations were thoroughly characterized regarding their soil properties and phosphorus status before the trial started (Table 1). The P status of the soils was analyzed according to the method for plant available P by the Association of German Agricultural Analytic and Research Institutes (VDLUFA). Phosphates were extracted with 100 mL solution of calcium acetate, calcium lactate and acetic acid buffered to pH 4.1 from 5 g air-dry soil followed by a photometric determination [28]. It is crucial to notice that all investigated soils showed levels of plant available P between 7.7 and 20.5 mg P/100g soil, therefore showing high to very high P availability according to the P-content-classes defined by the VDLUFA [29]. Overall, best agricultural management practice was followed, adapted to the individual agronomic demands of each location (e.g. Trichogramma treatment, herbicide application, etc.). The field season across locations ranged from 23rd of April to 29th of October 2019, sowing densities ranged from 8.8 to 10 plants/$m^2$, and plot sizes from 7.5 to 18 $m^2$ (for the latter only the middle rows were considered for grain harvest) according to the local standard practice (S3 Table).

## 2.3 | Phenotypic data

During the field season 2019, the following traits were assessed: plant height at up to four different developmental stages (PH, cm), ear height (EH, cm), days to anthesis (DTA, days after sowing, abbreviated as DAS), days to silking (DTS, days after sowing), anthesis-silking-interval

(ASI, days), grain dry matter content (GDM, %), grain yield (GY, t/ha), P grain concentration (Pconc, mg/kg), and P grain content (Pcont, kg/ha; calculated as GY*Pconc/1000). Details of how the traits were scored are provided in S4 Table. In case a trait was not measured at a location, the data were treated as not available (NA).

## 2.4 | Statistical analyses

First, we checked the quality of the phenotypic data of all traits on the single location level. The statistical model for this analysis was:

$$y_{ij} = \mu + g_i + r_j + \varepsilon_{ij}, \tag{1}$$

where $y_{ij}$ stands for the trait value of the $i$-th genotype in the $j$-th replicate; $\mu$ denotes the overall mean, $g_i$ the effect of the $i$-th genotype, $r_j$ the effect of the $j$-th replicate and $\varepsilon_{ij}$ the residual. Outlier detection was performed on the single location level applying the Bonferroni-Holm method [30].

In a second step, the analysis was performed across locations and the mixed model of the single location analysis was extended to the full model:

$$y_{ijkl} = \mu + t_i + g_j + l_k + (tg)_{ij} + (tl)_{ik} + (gl)_{jk} + (tgl)_{ijk} + r_{ikl} + \varepsilon_{ijkl} \tag{2}$$

where $y_{ijkl}$ stands for the trait value of the $j$-th genotype at the $k$-th location in the $l$-th replicate grown under the $i$-th starter fertilizer; $\mu$ denotes the overall mean, $t_i$ the effect of the $i$-th fertilizer treatment, $g_j$ the effect of the $j$-th genotype, $l_k$ the effect of the $k$-th location, $(tg)_{ij}$, $(tl)_{ik}$, $(gl)_{jk}$ represent the corresponding two-way interaction terms, $(tgl)_{ijk}$ the three-way interaction term, $r_{ikl}$ refers to the replication nested within the location and each starter fertilizer, and $\varepsilon_{ijkl}$ is the residual term. As for the single location analysis, all factors were treated as random to estimate the variance components except for the general mean and the starter fertilizer treatment which entered the model as a fixed factor for calculations across starter fertilizers. Significance of variance components was tested by likelihood ratio tests. Repeatabilities ($r^2$) and broad-sense heritabilities ($H^2$) respectively were calculated after the concept of the generalized heritability measure [31, 32] with $H^2 = 1 - A_{tt}/(2\sigma_g^2)$, where $H^2$ denotes the generalized heritability, $A_{tt}$ the average pairwise prediction error variance for the genotypic term, and $\sigma_g^2$ the genotypic variance estimate.

Best Linear Unbiased Estimates (BLUEs) were obtained for each of the investigated 20 hybrid varieties by considering the factor genotype as a fixed effect in the mixed model. All subsequent analyses were based on these BLUEs. Statistical analyses were performed with RStudio [33] and mixed model analyses were performed with ASReml-R [34]. In addition, the R-packages 'asremlPlus' [35] served for the calculation of information criteria for model selection, 'agricolae' [36] for the performance of significance tests, and 'multtest' [37] for outlier detection. Under R version 3.6.2 the R-packages 'ggpubr' [38], 'gplots' [39], and 'qgraph' [40] were used to produce plots.

## 3 | Results

### 3.1 | Response of traits to different starter fertilizers

The field trial underlying this study was based on five locations. Importantly, these can all be classified as having a high to very high P availability (Table 1). For all investigated traits, medium to very high repeatabilities were observed on the single location level. The lowest repeatabilities were found for grain yield with a minimum of 0.35, whereas grain dry matter content showed the highest values with a maximum of 0.98 (S5 Table). The phenotypic distributions and the

mean trait values revealed that if there was an effect of the starter fertilizer, it was usually the DAP treatment that exhibited this effect (Fig 1 and S2 Fig). Regarding the early plant height measurements, the control always showed the lowest mean, but only the DAP treatment resulted at youth stage in significantly taller plants than the control (Fig 1a). The response observed for plant height illustrates that the youth developmental stages are generally enhanced by the application of starter fertilizers. However, these differences diminished in the course of the field season and were not significant any more for the final plant height measurement. Nevertheless, ear height measurements resulted in significantly different means depending on the starter fertilizer (S2 Fig). The anthesis-silking-interval shortened from a mean of 0.65 days in the control and TSP to a mean of 0.40 days in the DAP treatment. These differences were statistically non-significant, but it must be noted that the ASI was very narrow for all 20 hybrids and only ranged between -1 and 4 days. Similarly, grain dry matter content was slightly higher in the DAP treatment with a mean of 69.40% compared to 68.15% in the control, indicating a faster maturity process in the treatments with starter fertilizer. These trends, even though they did not lead to significant differences, are in agreement with the significant differences observed for male and female flowering (S2 Fig). The DAP-fertilized varieties flowered significantly (p-value < 0.05) earlier (mean DTA = 83.03 DAS; mean DTS = 83.43 DAS) than the control (mean DTA = 84.80 DAS; mean DTS = 85.45 DAS), and also the TSP treatment (mean DTA = 83.75 DAS; mean DTS = 84.40 DAS) flowered approximately one day earlier than the control. For grain yield, the DAP treatment once again contrasted with the other treatments, yielding on average 13.21 t/ha, while the control showed a mean of 12.55 t/ha. In accordance with the results for grain yield, the highest P content was found for DAP with a mean of 31.61 kg/ha. P content showed only a significant difference (p-value < 0.05) between the DAP and CAN (mean Pcont = 29.60 kg/ha) starter fertilizer treatments. The P concentration of the grains, by contrast, showed no significant differences among the four treatments (S2 Fig).

## 3.2 | Relationships among traits dependent on the starter fertilization

The network plots visualized the correlations among the investigated traits dependent on the starter fertilizer (Fig 2). While there were differences, the general patterns remained the same. Grain yield, for instance, was always negatively correlated with P concentration, which can probably be attributed to the effect of dilution. Independent of the starter fertilizer, the early plant heights PH1, PH2, and PH3 (measured 53, 59, and 63 DAS, respectively) were closely related ($0.75 < r < 0.95$; p-values <0.01) but are no predictor for the final plant height (measured 94 DAS), nor grain yield. Another consistent triangle observed throughout the different treatments was the highly positive correlation between the male and female flowering times ($r > 0.9$, p-values < 0.001), which were always strongly negatively ($r > -0.87$, p-values < 0.01) correlated with grain dry matter (S3 Fig). Moreover, there was a significant positive association between the anthesis-silking-interval and the final plant height as well as between the P grain concentration and the P grain content.

We further analyzed the relationships between the maize kernel content of 16 chemical elements in the Co, TSP, and DAP treatment of the core location Hohenheim (S4 Fig). This revealed close positive correlations of phosphorus with magnesium, manganese, potassium, sulfur, and zinc.

## 3.3 | Identification of P sensitive and P stable genotypes across multiple locations

Having observed an effect of the starter fertilizer on some traits, the question arises whether the overall ranking of the varieties changes, i.e. whether there is a genotype-by-starter fertilizer

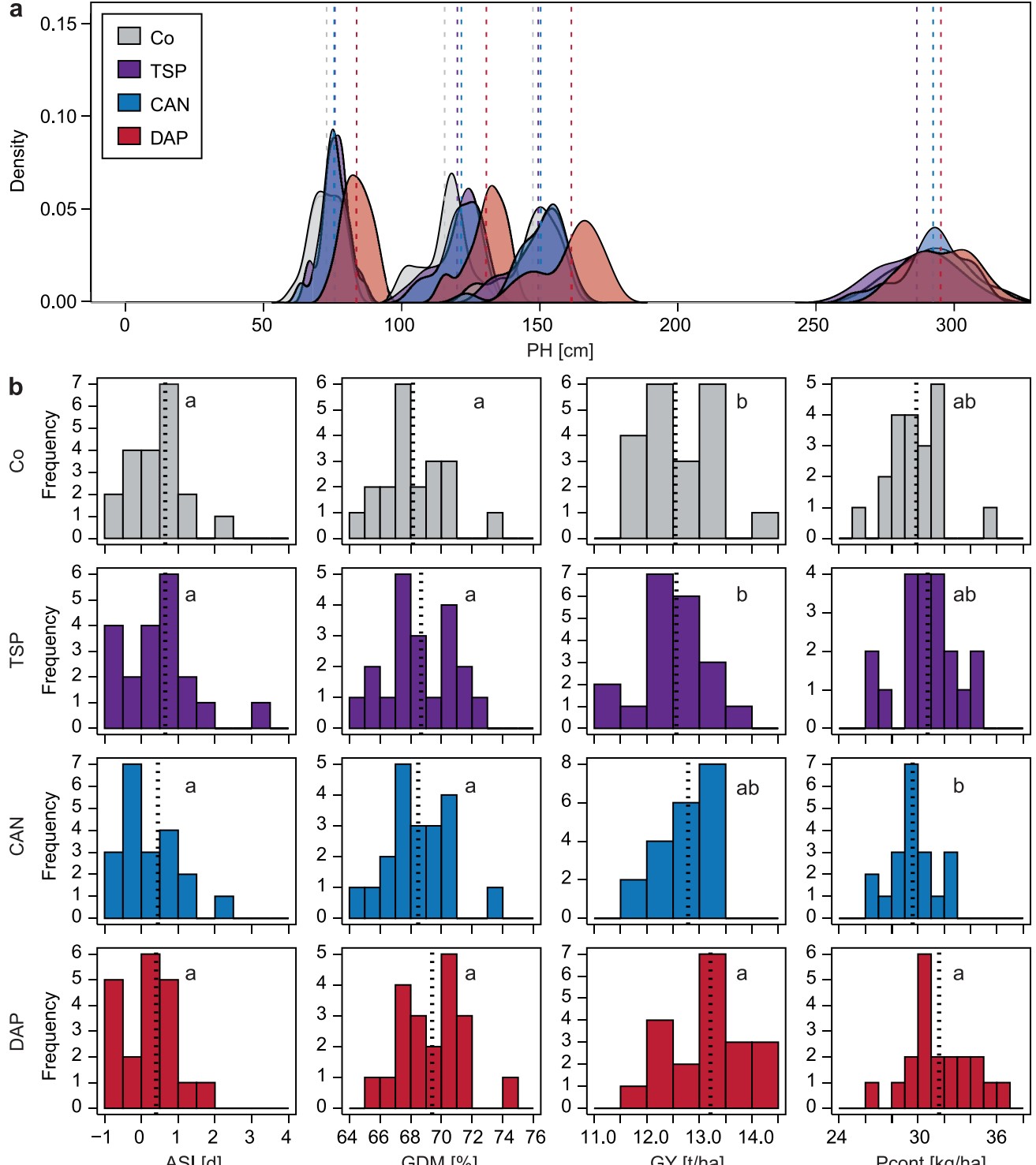

**Fig 1. Response of different traits to starter fertilizers.** Control (Co, grey), triple superphosphate (TSP, purple), calcium ammonium nitrate (CAN, blue), diammonium phosphate (DAP, red). (a) Density plots of plant height distributions at four different time points (PH1, PH2, PH3, PHfinal). (b) Histograms of anthesis-silking-interval (ASI), grain dry matter (GDM), grain yield (GY), and P content (Pcont). Different letters indicate significant (p-value < 0.05) differences between starter fertilizer means.

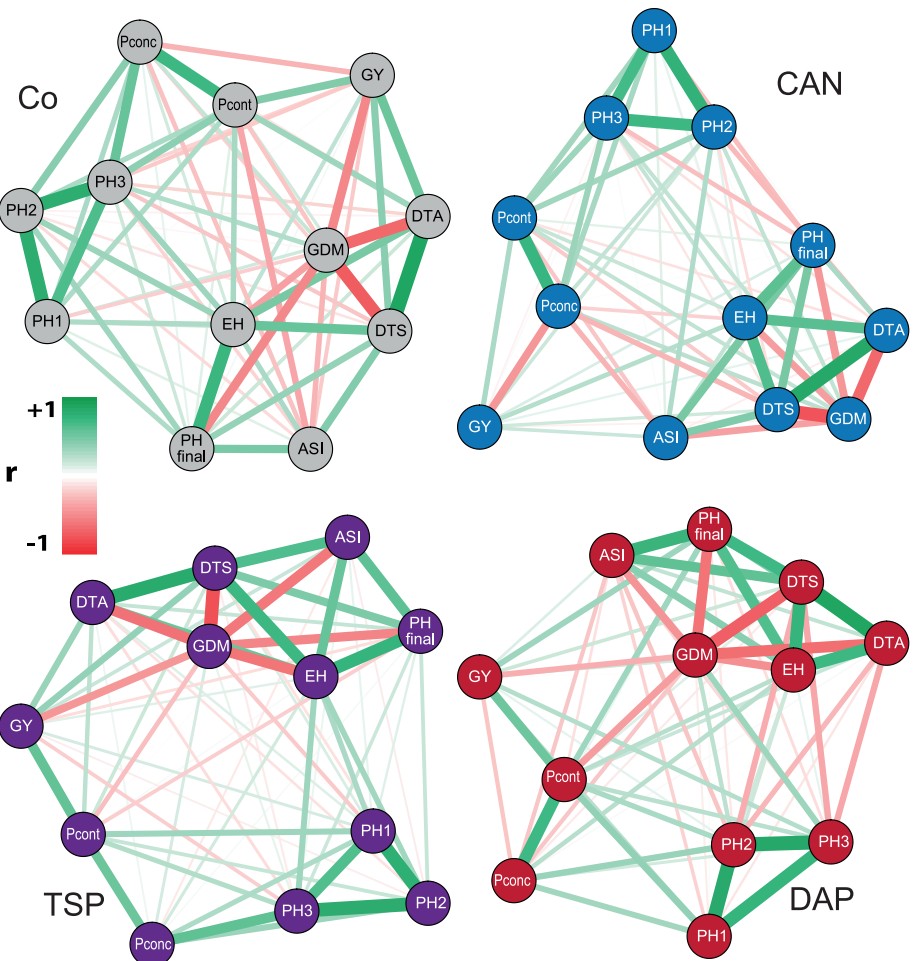

**Fig 2. Associations among the investigated traits dependent on the starter fertilizer.** Control (Co, grey), triple superphosphate (TSP, purple), calcium ammonium nitrate (CAN, blue), diammonium phosphate (DAP, red). Network plots showing anthesis-silking-interval (ASI), days to anthesis (DTA), days to silking (DTS), ear height (EH), grain dry matter content (GDM), grain yield (GY), P grain concentration (Pconc), P grain content (Pcont), and plant heights (PH1, PH2, PH3, PHfinal). Positive Pearson correlations (r) are indicated in green, negative Pearson correlations in red.

interaction. This knowledge is essential for the choice of variety by the farmer but also to choose appropriate conditions for genotype selection in breeding. The rank changes across the four different starter fertilizer treatments in Hohenheim demonstrated a rather parallel shift of the performance for grain yield and P content (Fig 3). The genotype-by-treatment interaction was non-significant for grain yield (p-value = 0.46). Thus, the best performing varieties under starter fertilizer application tend to be also among the best performing varieties in the control. Likewise, the trait P content signifying the removal of P from the field showed no significant genotype-by-treatment interaction (p-value = 0.30).

This observation, made for the core location Hohenheim, was confirmed in the series across locations, for which the genotype-by-starter fertilizer interaction in the analysis of variance was never significant for any observed trait (S6 Table). The heritabilities in the series were very high except for grain yield and the trait P content derived from it. This is likely due to the highly quantitative nature of grain yield, the rather small genotypic variation in this elite material and the observed strong genotype-by-location interaction. Again, the differentiation

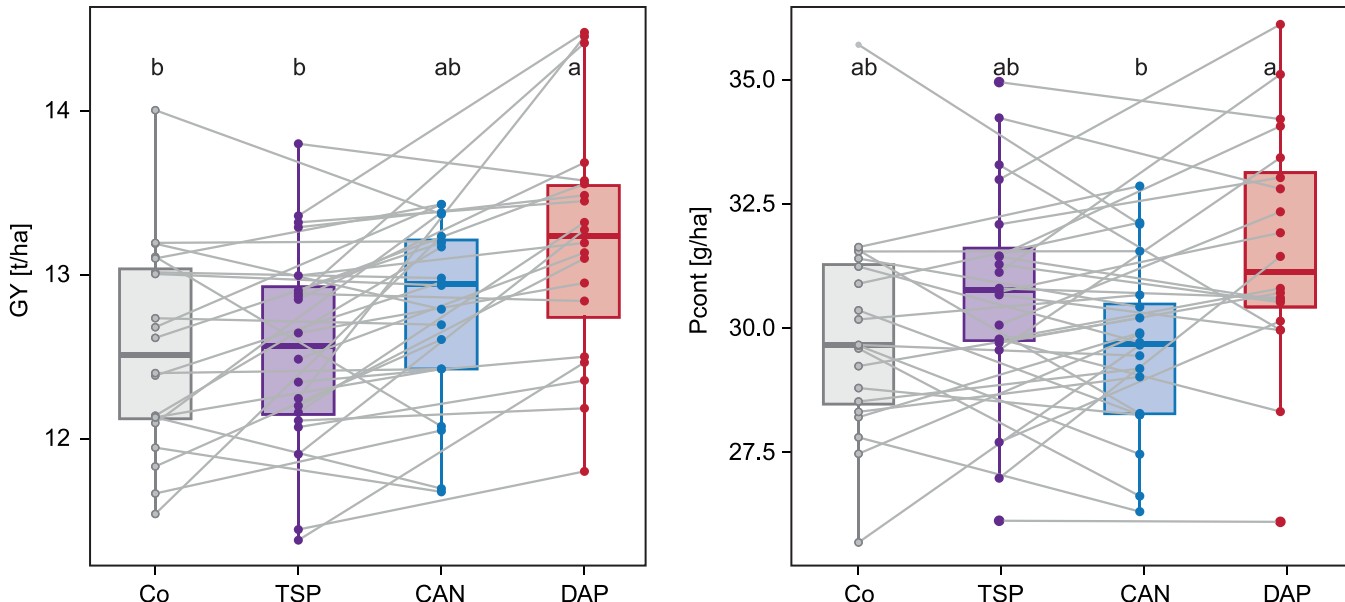

**Fig 3. Visualization of the genotype-by-starter fertilizer interaction.** Boxplots for grain yield (GY) and P grain content (Pcont) by starter fertilizers: control (Co, grey), triple superphosphate (TSP, purple), calcium ammonium nitrate (CAN, blue), diammonium phosphate (DAP, red). Interaction terms are visualized by connecting trait values of the same varieties with a grey line. Note that for the sake of clarity lines are only drawn between every other treatment. Different letters indicate significant (p-value < 0.05) differences between starter fertilizers means.

was more pronounced in the comparison between DAP and the control than between TSP and the control.

Interestingly, the origin of the variety in the sense of the eight different breeding companies, did not lead to a consistently similar behavior with regard to the investigated traits under different starter fertilizers (S5 Fig). From a practical farming point of view, the most interesting question is how the different varieties perform with regard to grain yield under control and starter fertilizer conditions. Therefore, relative grain yields under TSP or DAP starter fertilizer were plotted against the relative grain yield of the control for each location (Fig 4). Varieties in quadrant I (highlighted upper right) of the plot can be defined as stably above- average-yielding P-utilizers, whereas varieties in quadrant III (bottom left) are relatively low-yielding independent of their P-supply. Varieties located in the quadrants II and IV can be considered P-sensitive genotypes, as they will show above-average yields with starter fertilizer but not in the control or vice versa, respectively. In line with the small genotype-by-treatment interaction, most varieties showed either below-average or above-average yield performance no matter which starter fertilizer they were grown under. By comparing the relative performance of all varieties in each treatment-location-combination, we identified the consistently best varieties. For the TSP-series, three varieties were in quadrant I at each location (AGROPOLIS_AM, AMAVERITAS_AM, WALTERINIO_KWS) and for the DAP-series also three varieties (AGROPOLIS_AM, FIGARO_KWS, SY_TALISMAN). Only the variety AGROPOLIS_AM was in this high-yielding quadrant at each location-treatment combination.

From a breeding point of view, it is also interesting to identify the most P-independent genotypes. While the analysis of the relative performance in the control and the starter fertilizer treatments already provided some indication to this, genotypes may be above average for both treatments, but still show a substantial reduction in grain yield when the starter fertilizer is omitted. The most interesting candidates are those showing the least reduction between starter fertilizer and control, while at the same time having a high yield. We therefore analyzed

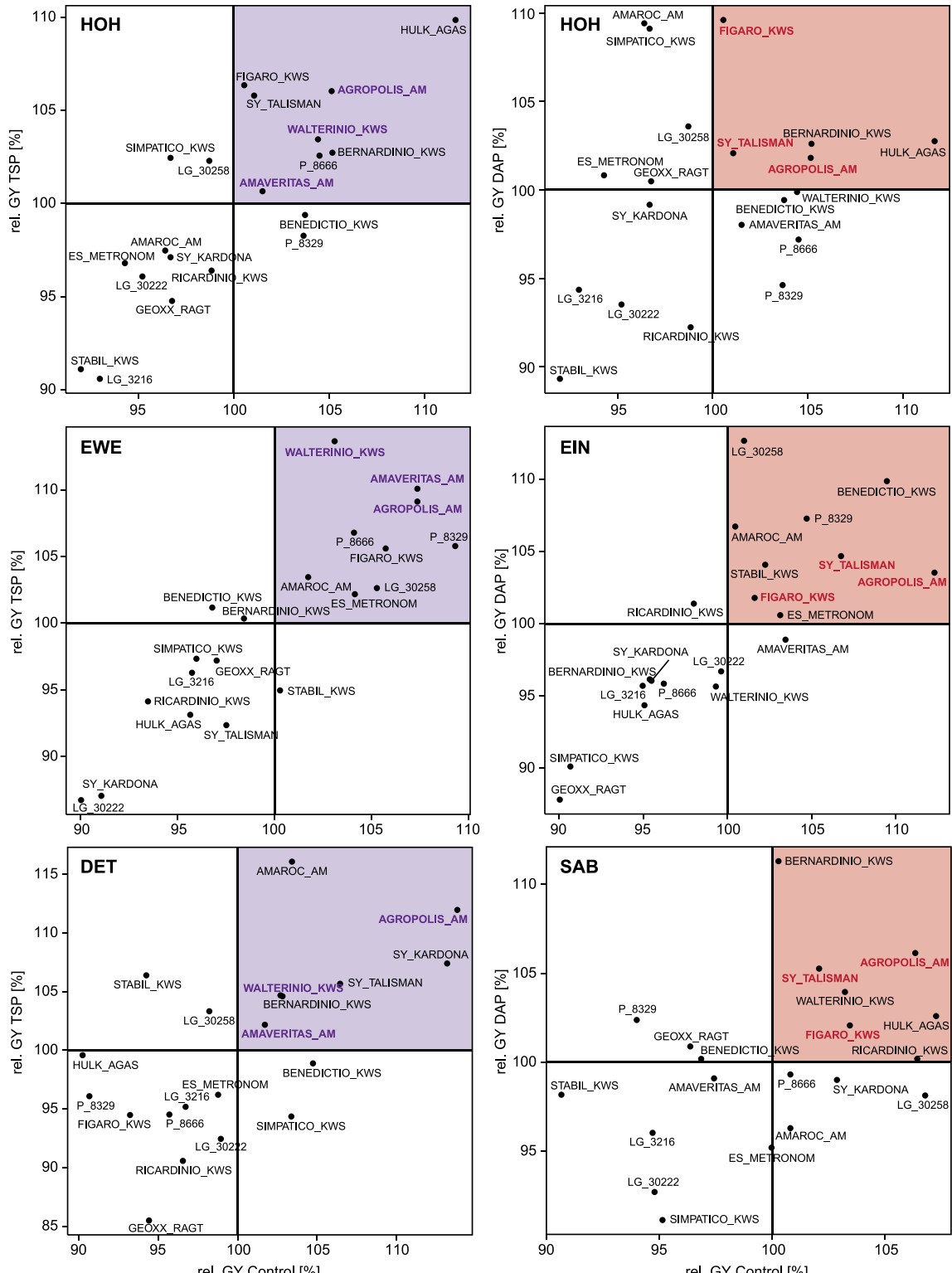

**Fig 4. Scatterplots of each starter fertilizer-location combination.** Relative grain yield (rel. GY [%]) in the control vs. triple superphosphate (TSP, purple) or vs. diammonium phosphate (DAP, red), respectively. Quadrants are counted starting from the highlighted one as I (stably above- average-yielding P-utilizers) in an anti-clockwise manner to IV (II and IV indicate P-sensitivity). The varieties which in all three locations were located in Quadrant I are highlighted in bold and color. Locations are abbreviated as follows: Hohenheim (HOH), Eckartsweier (EWE), Dettingen (DET), Einbeck (EIN), and Saerbeck (SAB).

| Variety | ΔGY TSP-Co [%] | ΔPcont TSP-Co [%] | ΔGY DAP-Co [%] | ΔPcont DAP-Co [%] |
|---|---|---|---|---|
| AGROPOLIS_AM | -1,63 | -8,24 | 0,15 | -7,21 [$] |
| AMAROC_AM | -6,01 | -6,23 | -8,30 | -18,30[$] |
| AMAVERITAS_AM | -1,72 | -0,99 | -1,43 | 3,14 [$] |
| BENEDICTIO_KWS | 1,03 | 3,03 | -3,12 | -2,47 [$] |
| BERNARDINIO_KWS | -1,50 | -5,33 | -6,13 | -8,64 [$] |
| ES_METRONOM | -0,49 | -3,04 | -3,59 | -11,63[$] |
| FIGARO_KWS | -3,37 | -6,88 | -6,22 | -19,04[$] |
| Geoxx_RAGT | 2,02 | 0,72 | -6,37 | -7,58 [$] |
| HULK_AGAS | -2,74 | -2,96 | 1,52 | 4,38 [$] |
| LG_30222 | 1,54 | 8,03 | -1,23 | -4,28 [$] |
| LG_30258 | -3,19 | 6,03 | -5,57 | -6,24 [$] |
| LG_3216 | 0,15 | -1,76 | -4,62 | -9,90 [$] |
| P_8329 | 0,17 | -0,34 | -3,59 | -5,37 [$] |
| P_8666 | -0,78 | -1,62 | -0,09 | 4,48 [$] |
| RICARDINIO_KWS | 1,74 | 4,22 | 0,21 | 1,33 [$] |
| SIMPATICO_KWS | -0,53 | -0,36 | -6,65 | -16,88[$] |
| STABIL_KWS | -2,11 | 0,65 | -5,49 | 5,25 [$] |
| SY_KARDONA | 1,83 | 1,45 | -3,20 | -3,74 [$] |
| SY_TALISMAN | -0,98 | -0,40 | -4,22 | 2,24 [$] |
| WALTERINIO_KWS | -4,33 | -7,77 | -1,04 | -2,81 [$] |

[$]only assessed for the location HOH

**Fig 5. Relative differences for each hybrid variety.** Grain yield (GY) and P content (Pcont) are based on best linear unbiased estimators (BLUEs) across locations between the fertilized treatment (purple: triple superphosphate (TSP); red: diammonium phosphate (DAP)) compared to the control. The darker the coloration, the stronger the reduction.

for each variety the reduction in grain yield and P concentration between the DAP or TSP starter fertilizer and the control across locations. This revealed the trend that if grain yield is strongly reduced without starter fertilizer, this generally goes in line with a reduced P content (Fig 5), while the P concentration does not significantly (p-value < 0.05) differ among the starter fertilizer treatments. Also, varieties that performed consistently above-average, like for instance AGROPOLIS_AM, can nonetheless show proportionally strong reductions without starter fertilization.

## 4 | Discussion

This study was performed to investigate the effect of different starter fertilizers on current maize hybrids in Germany. The application of starter fertilizer is common practice in maize cultivation, but whether there are genotypes for which it can be omitted has not been thoroughly investigated yet. Now, however, that the application of fertilizers in agriculture, including the associated negative environmental effects, has come under focus and is becoming increasingly restricted, this question has gained social and political relevance and warrants scientific answers. We therefore screened the top 20 market leaders of German maize hybrid varieties in five locations under DAP (N+P), TSP (P), CAN (N) and Co (no) starter fertilization to assess potential P fertilizer reductions, evaluate the underlying genotypic components, identify superior genotypes regarding P supply, and draw conclusions for practical maize breeding.

### 4.1 | P as starter fertilizer and its interaction with other plant nutrients

The importance of a balanced nutrient management and specifically the role of nitrogen in fertilization is well known [5]. The results obtained in our study suggest a synergetic effect of a co-starter-fertilization of ammonium and phosphate. This combination had the strongest effect on early plant height measurements and thus youth development, and led to yield increases compared to the control treatment (Figs 1 and 3). Even though yield clearly represents the most important trait for the farmer, a successful youth development in farmers' fields is not to be underestimated. Interestingly, DAP starter fertilization also resulted in a slight increase of the P content, thus the amount of P that was successfully taken up by the plants and that is eventually removed from the field. This effect was extensively observed in former studies [25] and can be explained by the acidification ammonium causes in the soil, which enhances P uptake [25, 41]. More precisely, local ammonium supply stimulates the extension of the root system [42], which is caused by the accumulation of the plant hormone auxin [43]. In general, the soil conditions e.g. pH, anion and metal concentrations [21] as well as the effect of the previous crop and the crop rotation strongly impact the bioavailability of P in the rhizosphere [44]. In which combination P is given to the maize plants seems to be crucial for its successful conversion in the plant. The CAN treatment does not appear to unleash the available P in the soil, which is likely due to its lower acidifying potential compared to DAP [45].

Another interesting aspect when talking about the relationship of P with other plant nutrients is the consistent positive relationship of phosphorus with manganese, magnesium, potassium, sulfur, and zinc observed across all starter fertilizers (S4 Fig), which prevails also when looked at each starter fertilizer separately. Previous studies confirmed that potassium, manganese and magnesium were highly positively correlated with P in maize grains [46]. This underlines the need to check for example for a sufficiently high magnesium status of the fields, which—in case it is limited—should be applied as an efficient fertilizer combination as customary in trade. Taken together, these observations highlight the importance to keep in mind other nutrients besides P that promote maize youth development. When working on the improvement of phosphate-use-efficiency, we also have to consider the overall nutritional status in the soils, also with regard to suitable co-fertilization strategies and even planning of crop rotations, which determine the whole cropping system.

### 4.2 | Potential for optimizing the P balance on well-supplied soils

It is paramount to emphasize that all trial locations showed no P deficiency of the soils. By contrast, all soils can be classified as rich to very rich soils with regard to P availability (Table 1). For all further considerations, we therefore have to keep in mind that the starter fertilizer treatments took place on fields with an overall very good nutrient availability. While some parts of the world are challenged with P-deficient soils, in Germany this situation of well-supplied soils is rather the rule than the exception [18, 47]. Different studies underlined that current P stocks in the soils in Europe allow for sufficient P supply of the crops for several years with zero fertilization [18, 48]. In practical farming, the application of starter fertilizers is often simply conditioned by the availability of the corresponding sowing technique. Our results showed that only the combination of ammonium and phosphate (DAP) as starter fertilization resulted in significantly higher grain yields by on average +0.67 t/ha at the core location Hohenheim (Figs 1b and 3), as well as on average +0.4 t/ha over multiple locations, which corresponds to an increase of +5.34% and +3.6%, respectively (S6 Table). By contrast, the yield increase using only phosphate (TSP) as starter fertilizer only amounted to +1.2% across multiple locations. Thus, the commonly applied combination of ammonium and phosphate as starter fertilizer does have a positive effect on maize yield, at least on average across all 20 hybrid varieties.

Notably, however, performance without starter fertilizer has not been a breeding goal to date. Hence, there is a certain potential to reduce or omit P starter fertilizers and thereby gain lee-way in the farm nutrient balance, even if this may come at the price of potential minor reductions in grain yield. At the same time, P surpluses on a farm but also on a regional level are or will be in the future restricted and fined by law, which makes it worthwhile for the farmer to thoroughly weigh additional fertilizer versus additional yield. We conclude that meaningful phosphate-use-efficiency in the context of well-saturated soils should be defined ideally as only minor yield reductions without extra P fertilization given as starter fertilizer.

### 4.3 | How to breed for phosphate-use-efficiency?

Two aspects determine if breeding of maize hybrids with a reduced need for starter fertilizer is possible and how it can be pursued. First, we need genetic variation regarding the response to reduced or no starter fertilizer, so that lines with no or only a minimal reduction in growth and yield can be selected. If so, the genotype-by-starter fertilizer interaction will determine under which conditions selection should be performed.

Our results show that there is variation regarding the response to starter fertilizer and thus the potential to omit it. Identifying and selecting genotypes that are high-yielding and maintain above-average performance regardless of the starter fertilization, is thus possible and can be considered a meaningful goal for breeding in Germany. As described in the literature [49], we also observed a shift of flowering dates due to the different starter fertilizers (S2 Fig). Generally speaking, the better the soil is supplied with P, the earlier the flowering takes place. In our case, however, this shift only amounted to less than one day and is of no practical relevance.

We observed neither a significant genotype-by-starter fertilizer interaction for the trait grain yield nor for P content (Fig 3). This suggests that breeders can select P-efficient lines independent of the soil P-status since generally the best genotypes perform the best no matter with or without starter fertilization. Nonetheless, further research is required to investigate whether this also holds true for soils with lower P availability than investigated in this study. With the expected restrictions for P fertilizer inputs ahead, breeders should still target to select under no P starter fertilizers conditions for obtaining better adapted material with regard to phosphate-use-efficiency.

For breeding purposes, more genotypes should be screened in more locations, including poorer P availability classes, and more importantly, the trials should be carried out in more years. It is known from other studies that the effect of starter fertilizers is extremely dependent on the environment [24, 50] and on the year [9]. As pointed out, the early phase of the field season in 2019 was extraordinary wet and cold (S1 Fig). The application of starter fertilizer may buffer against such adverse events and thus provide a kind of insurance for the farmer. This potential positive effect must be weighed against legal regulations restricting fertilizer use per farm. Obviously, the availability of varieties that do not require this external buffer in the form of starter fertilizer, but have a strong youth development and can cope with a certain level of abiotic stress genetically, would be an important means to reduce P input in our agricultural systems. More and more seed treatments that enhance the mobilization of P in the soils are currently entering the market and show additional ways of how a sustainable optimized P balance can be achieved in the future.

### 5 | Conclusions

Our study revealed that starter fertilizer treatments have a rather limited effect on grain yield but mainly show a positive effect on the youth development of maize. Breeding for phosphate-

use-efficiency in the context of well-supplied soils, as present for example in Germany, should focus on genotypes that maintain high absolute grain yields even with a reduction of P inputs to zero. Selection of such phosphate-use-efficient varieties appears possible without taking the P level of the soil into account, since no substantial genotype-by-starter fertilizer interaction is expected under the P-rich soil conditions to be mostly found in Germany. In order to fulfill the clear social and political will of reducing fertilizers, plant breeding should contribute its part and provide varieties that allow the desired reduction of fertilizers without major financial disadvantages for the farmers.

## Supporting information

**S1 Table. Detailed description of hybrid varieties.** Information of all 20 hybrids investigated in the field season 2019, including the breeding company, maturity (FAO groups go from early 170–220 to late 300–350), the main utilization ('B' denoting biogas, 'CCM' corn-cob-mix, 'G' grain and 'S' silage), and the companies standard seed treatment. The year of registration is given according to the federal plant variety office.
(PDF)

**S2 Table. Starter fertilizer-location combinations.** Control (Co), triple superphosphate (TSP), calcium ammonium nitrate (CAN), diammonium phosphate (DAP). In brackets the nitrogen and phosphorus content are given in percent.
(PDF)

**S3 Table. Maize cultivation parameters.** Given for the field season 2019 in each location.
(PDF)

**S4 Table. Detailed description of trait assessments.** Same methods were applied for all locations.
(PDF)

**S5 Table. Repeatabilities in the single locations.** Traits are abbreviated as follows: Plant height <55 days after sowing (DAS)(PH1), Plant height 56–60 DAS (PH2), Plant height 61–65 DAS (PH3), Plant height 66–70 DAS (PH4), Plant height 71–75 DAS (PH5), Plant height >75 DAS (PHfinal), ear height (EH), all measured in cm; days to anthesis (DTA) and days to silking (DTS), indicated in DAS; anthesis-silking-interval (ASI) in days; grain dry matter content (GDM) in percent; grain yield (GY) in tons dry matter/ha; Phosphorus grain concentration (P conc) measured with X-ray fluorescence in mg P/kg dry matter; and Phosphorus grain content (P cont) in kg P/ha. Control (Co, grey), starter fertilizers: triple superphosphate (TSP, purple), calcium ammonium nitrate (CAN, blue), diammonium phosphate (DAP, red).
(PDF)

**S6 Table. Summary of the statistical analyses in the series.** (i) Control (Co) vs. triple superphosphate (TSP) and (ii) Control (Co) vs. diammonium phosphate (DAP): Values are given for within each starter fertilizer treatment (indicated with Co, TSP, DAP, respectively) as well as across both starter fertilizer treatments. Minimum (Min), Mean, and Maximum (Max) is given based on the best linear unbiased estimators (BLUEs). $\sigma^2_g$ denotes the genotypic variance, $\sigma^2_l$ the location variance, $\sigma^2_{gxt}$ the genotype-by-treatment-interaction variance, $\sigma^2_{gxtxl}$ the genotype-by-treatment-by-location-interaction variance, $\sigma^2_e$, the error variance, and $H^2$ the broad-sense heritability. Traits are abbreviated as follows: plant height at BBCH stage ~ V4 (PH early), plant height at BBCH stage > R1 (PH late), ear height (EH), days to silking (DTS) given in in days after sowing (DAS), grain dry matter (GDM), grain yield (GY), phosphorus grain concentration (P grain conc), and phosphorus grain content (P cont). Significance levels

are shown as '*' (p-value < 0.05), '**' (p-value < 0.01), '***' (p-value < 0.001). All values are based on three locations (Co-TSP: Hohenheim, Eckartsweier, Dettingen; Co-DAP: Hohenheim, Einbeck, Saerbeck), except for P grain conc and P cont in the DAP series.
(PDF)

**S7 Table. Raw data of hybrid trial.**
(XLSX)

**S1 Fig. Climograph and soil temperatures at the location Hohenheim.** Daily precipitation rates [mm/d] and mean temperatures [˚C] of the year 2019. The dates of plant height measurements during the field season are indicated with dark green arrows, the period of the field season with a light green arrow. Soil temperatures at 2 cm, 20 cm, and 200 cm are shown in brown colors in the plot below.
(PDF)

**S2 Fig. Histograms of specific traits.** The traits final plant height (PHfinal), ear height (EH), days to anthesis (DTA), days to silking (DTS), and P concentration (Pconc) are depicted. Different letters indicate significant (p-value < 0.05) differences between starter fertilizers means. Starter fertilizers are abbreviated as Control (Co), triple superphosphate (TSP), calcium ammonium nitrate (CAN), diammonium phosphate (DAP).
(PDF)

**S3 Fig. Correlation matrices of all investigated traits.** Separated by starter fertilizers: control (Co), triple superphosphate (TSP), calcium ammonium nitrate (CAN), diammonium phosphate (DAP). Anthesis-silking-interval (ASI [d]), days to anthesis (DTA [d]), days to silking (DTS [d]), ear height (EH [cm]), grain dry matter content (GDM [%]), grain yield (GY [t/ha]), P grain concentration (Pconc [mg/kg]), P grain content (Pcont [kg/ha]), and plant heights (PH1, PH2, PH3, PHfinal [cm]). Red indicates negative correlations between traits, green positive correlations. Significance levels are shown as '.' (p-value < 0.1), '*' (p-value < 0.05), '**' (p-value < 0.01), '***' (p-value < 0.001).
(PDF)

**S4 Fig. Network plot among 16 chemical elements.** 120 grain samples of the core location HOH were analyzed, independent of starter fertilizer treatments. Positive Pearson correlations (r) are indicated in green, negative Pearson correlations in red.
(PDF)

**S5 Fig. Heatmaps of all 20 maize hybrids and the investigated traits.** Separated by starter fertilizers (control (Co), triple superphosphate (TSP), calcium ammonium nitrate (CAN), diammonium phosphate (DAP)): anthesis-silking-interval (ASI [d]), days to anthesis (DTA [d]), days to silking (DTS [d]), ear height (EH [cm]), grain dry matter (GDM [%]), grain yield (GY [t/ha]), P concentration (Pconc [mg/kg]), P content (Pcont [kg/ha]), and plant heights (PH1, PH2, PH3, PHfinal [cm]). Dark red indicates maximum, light yellow minimum trait values.
(PDF)

## Acknowledgments

We are grateful to Franz Josef Mauch, who managed the core location Hohenheim and Dettingen in an excellent way. We also thank the experimental station Eckartsweier, namely Regina Bauer and Christiane Maus, for conducting the XRF-measurements. Moreover, we thank Sidhant Chaudhary for his support in collecting the phenotypic data during his master's thesis.

All breeding companies that took part in the trial we want to give credit for providing the seeds as well as for their time and efforts to generate and share the data. Especially, we would like to acknowledge Daniel Wolper (KWS), Thomas Lachenmayer (RAGT), and Nic Boerboom (DSV). Last but not least, we are very much obliged to Walter Schmidt for his highly appreciated advice in choosing the plant material of this study.

## Author Contributions

**Conceptualization:** Thea Mi Weiß, Willmar L. Leiser, Tobias Würschum.

**Data curation:** Thea Mi Weiß.

**Formal analysis:** Thea Mi Weiß.

**Investigation:** Thea Mi Weiß, Alice-J. Reineke.

**Methodology:** Dongdong Li, Tobias Würschum.

**Project administration:** Thea Mi Weiß.

**Validation:** Volker Hahn.

**Visualization:** Thea Mi Weiß, Dongdong Li.

**Writing – original draft:** Thea Mi Weiß.

**Writing – review & editing:** Thea Mi Weiß, Willmar L. Leiser, Alice-J. Reineke, Dongdong Li, Wenxin Liu, Volker Hahn, Tobias Würschum.

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
