## [Decision Letter · Decision Letter 0]

17 Mar 2021

PONE-D-20-36545

Optimizing the P balance: How do modern maize hybrids react to different starter fertilizers?

PLOS ONE

Dear Dr. Weiß,

Thank you for submitting your manuscript to PLOS ONE. After careful consideration, we feel that it has merit but does not fully meet PLOS ONE’s publication criteria as it currently stands. Therefore, we invite you to submit a revised version of the manuscript that addresses the points raised during the review process.

Address all queries raised by all four reviewers and revise the manuscript at the earliest

We look forward to receiving your revised manuscript.

Kind regards,

Kandasamy Ulaganathan

Academic Editor

PLOS ONE

Journal Requirements:

'The funders had no role in study design, data collection and analysis, decision to publish, or preparation of the manuscript'

Additional Editor Comments (if provided):

Reviewers' comments:

Reviewer's Responses to Questions

**Comments to the Author**

1. Is the manuscript technically sound, and do the data support the conclusions?

Reviewer #1: Yes

Reviewer #2: No

Reviewer #3: Partly

Reviewer #4: Yes

2. Has the statistical analysis been performed appropriately and rigorously? 

Reviewer #1: Yes

Reviewer #2: No

Reviewer #3: Yes

Reviewer #4: Yes

3. Have the authors made all data underlying the findings in their manuscript fully available?

Reviewer #1: Yes

Reviewer #2: No

Reviewer #3: Yes

Reviewer #4: Yes

4. Is the manuscript presented in an intelligible fashion and written in standard English?

Reviewer #1: Yes

Reviewer #2: Yes

Reviewer #3: Yes

Reviewer #4: Yes

5. Review Comments to the Author

Reviewer #1: 1. In the abstract, authors claimed yield increase of on average +0.67 t/ha (+5.34 %) compared to the control with DAP while in the discussion section they are claiming on average +0.4 t/ha (+3.6 %) higher yield over multiple locations. Please check the same.

2. Line no. 73-73:

In many cases however, the abundant P is fixed by minerals and therefore not fully plant available.

May be written as:

In many cases however, the abundant P is fixed by minerals and therefore not fully available for plants.

3. Line no. 100:

We applied four different starter fertilizers, namely….

Since authors have used three starter fertilizers and one control..So this may be written as:

We applied four treatments i.e. control and three different starter fertilizers…

4. Figure S5 and S6 cited before S4 in text. Please re-arrange figures to cite them sequentially.

Reviewer #2: 1. Insufficient experimentation: The study involving three fertilizer treatments is conducted at one location and the data is of one year. Other locations do not have all the treatment combinations.

2. The title mentions optimizing P balance and the experiment conducted on P sufficient soils will not give reliable results. Hence, Also G x fertilizer effects cannot be studied from the experimentation and the discussion done on this point is not valid.

3. Abstract is not clear

4. ANOVA table not presented. How the experiments analyzed from all locations not given. Also locations are not discussed in material and methods and other sections.

5. Manuscript has to be improved for clarity in several places and well supported with relevant tables and figures.

6. Authors need to follow standard notations ex. Field seasons, youth etc. used in paper

Reviewer #3: 1. The study undertaken was one of the most important aspect which would try to address the issue of development of ‘P’ use efficient genotypes in the context of ‘P’ being scarce resource and expected to exhaust in near future. However, based on the following points, the manuscript require major revision in the light of below observations.

2. The authors have used TSP, DAP and CAN to know the difference between different starter fertilizers but authors have not mentioned or studied the effect of other nutrients like nitrogen on ‘P’ utilization. Because balanced use of fertilizers is advocated for efficient use of fertilizers.

3. Authors are requested to justify the validity of the results in the practical situation of maize cultivation where the soil contains different nutrients in varying amounts, how does the results of the present experiment would apply in practical situation.

4. The experimental results are more of an agronomic perspective rather than pure breeding perspective thus the authors are requested to re-orient the results in breeding perspective.

5. In addition, the following points may be considered while improving the manuscript

a. Youth development is the new terminology used for early development phase it may be clarified by giving leaf stage (like V3/V4/V5/V6) for larger benefit of the readers.

b. The references namely 1,17,27,28,29 may also be translated into English for the benefit of the readers to know at least the title of the paper.

c. Sentence in line 73-74 may be revised

d. Reference 23, does apply to maize please clarify to readers

e. Reference 24 (line 436 to 438) t is better to use symbol ‘×’ instead of letter ‘x’; is reference 24 relevant, please have a re-look.

f. Line 228 may need to be re-looked at for spelling correction of there-of

Reviewer #4: Comments on MS PONE-D-20-36545 entitled “Optimizing the P balance: How do modern maize hybrids react to different starter fertilizers?”

The study is on optimizing P balance in maize hybrids and how they react to starter fertilizers, the authors have studies how to reduce starter P for maize hybrids. The study has come out with some novel findings on.

Line 61 – correct 1980ies to 1980s

The introduction is written well, the hypothesis is stated well

The materials section is well written and the experiment is well planned and all the details of the experiment is given in a good way in the section

The results section is written well with all the results obtained given in the section

The discussion section brings out the reasons for the results obtained in a good way

The MS can be accepted after minor revision

6. PLOS authors have the option to publish the peer review history of their article (what does this mean?). If published, this will include your full peer review and any attached files.

Reviewer #1: **Yes: **Sujay Rakshit

Reviewer #2: No

Reviewer #3: No

Reviewer #4: No

---

## [Author Response · Author response to Decision Letter 0]

23 Mar 2021

Please find rebuttal letter as well as the answers to the reviewers uploaded along with the manuscript.

---

## [Decision Letter · Decision Letter 1]

8 Apr 2021

Optimizing the P balance: How do modern maize hybrids react to different starter fertilizers?

PONE-D-20-36545R1

Dear Dr. Weiß,

We’re pleased to inform you that your manuscript has been judged scientifically suitable for publication and will be formally accepted for publication once it meets all outstanding technical requirements.

Kind regards,

Kandasamy Ulaganathan

Academic Editor

PLOS ONE

Additional Editor Comments (optional):

Reviewers' comments:

Reviewer's Responses to Questions

**Comments to the Author**

1. If the authors have adequately addressed your comments raised in a previous round of review and you feel that this manuscript is now acceptable for publication, you may indicate that here to bypass the “Comments to the Author” section, enter your conflict of interest statement in the “Confidential to Editor” section, and submit your "Accept" recommendation.

Reviewer #3: All comments have been addressed

Reviewer #4: All comments have been addressed

2. Is the manuscript technically sound, and do the data support the conclusions?

Reviewer #3: Yes

Reviewer #4: Yes

3. Has the statistical analysis been performed appropriately and rigorously? 

Reviewer #3: Yes

Reviewer #4: Yes

4. Have the authors made all data underlying the findings in their manuscript fully available?

Reviewer #3: Yes

Reviewer #4: Yes

5. Is the manuscript presented in an intelligible fashion and written in standard English?

Reviewer #3: Yes

Reviewer #4: Yes

6. Review Comments to the Author

Reviewer #3: The responses given by the authors against the comments are found satisfactory, hence the manuscript is recommended for consideration in the publication

Reviewer #4: The authors have revised the MS according to the suggestions of the reviewers

the MS can be accepted in its present form

7. PLOS authors have the option to publish the peer review history of their article (what does this mean?). If published, this will include your full peer review and any attached files.

Reviewer #3: No

Reviewer #4: No

---

## [Editor Report · Acceptance letter]

13 Apr 2021

PONE-D-20-36545R1 

Optimizing the P balance: How do modern maize hybrids react to different starter fertilizers? 

Dear Dr. Weiß:

I'm pleased to inform you that your manuscript has been deemed suitable for publication in PLOS ONE. Congratulations! Your manuscript is now with our production department. 

Kind regards, 

on behalf of

Dr. Kandasamy Ulaganathan 

Academic Editor

PLOS ONE